# Initializing and Retrofitting Key-Value Adaptors for Traceable Model Editing

## Abstract

As the insight of knowledge storage in language models deepens, the ability to perform CRUD (Create, Read, Update, Delete) operations on language models becomes increasingly indispensable for satisfying the demands of managing rapidly updating knowledge. Considering the high cost of fine-tuning language models, model editing methods with low cost are usually required to manipulate models' knowledge. The evidence suggests that modules carrying knowledge in a Transformer module are primarily the MLP blocks, thus we propose **iReVa**, a method that explicitly initializes and retrofits key-value pairs into MLP blocks to construct a new mapping of a piece of knowledge without damaging the irrelevant knowledge. In comparison to existing methods, iReVa reveals better interpretability and a stronger capacity for carrying traceable edits. Experiment results on a series of GPT series models show our prominent performance on edit success and generalization without influencing specificity. We also made the first attempt to conduct a knowledge withdrawal test of iReVa. Our codes are available at this website.

## 1 Introduction

Language Models (LMs) Brown et al. (2020) are becoming imperative tools for consulting in real-world scenarios. One significant reason for the prevalence of LMs is their ability to answer factoid questions. For example, when we ask an LM with the question "*Who is president of America ?*", it returns the answer "*Joe Biden*". Even though a mass amount of knowledge is stored in the LMs, we still face the issue of out-of-date and missing knowledge Petroni et al. (2019); Jiang et al. (2020). Alternatively, some knowledge may change over years and some domain-specific knowledge may be absent from the LMs.

To bridge the gap, the task of model editing is introduced to *edit* the knowledge in LMs, which targets at modifying the parameters of LMs and inject certain knowledge to them Zhang et al. (2024). The difficulty of this task lies in the manipulation to the LMs, where the knowledge is implicitly stored in dense vectors. A naive solution to model editing is fine-tuning an LM with the new knowledge, whereas the cost is climbing with the surging size of LMs. More recent studies propose to directly update the models' weights in mastery phase Jayashri & Kalaiselvi (2018); Bruner (1960) via either teaching a hyper-network to learn the change of the weights or locating-then-editing knowledge neurons Cao et al. (2021); Mitchell et al. (2022a); Meng et al. (2023a;b). While the editing methods above are efficient in updating knowledge in LMs, they encounter the difficulties of differentiating the existing and new knowledge, which makes the editing hard to control. Methods like life-long model editing Hartvigsen et al. (2023), MELO Yu et al. (2023), and T-Patcher Huang et al. (2023) propose to learn the representation for new knowledge and merge this information with the original models.

However, these methods still conform to the paradigm of learning the batch edit Huang et al. (2023); Hase et al. (2021) as a whole without modeling edit parameters in a traceable way, which can not conform the edit success to each edit and have a lack interpretability to the editing. In contrast, we propose a method of **I**nitializing and **R**etrofitting KE**y**-**V**alue **A**daptors (**iReVa**), an editing method that inserts a key-value adaptor to indicate the mapping of an edit data pair and further retrofit the adaptor with multiple objectives. Moreover, to prevent the unnecessary change to the irrelevant knowledge, we elaborately design activation mechanism for the knowledge neurons. Experimental results on series of GPT-like models show that iReVa is able to outperform the SOTA results by

around 9% and 6% average score improvement on zsRE-10K and PARAREL-10K, respectively. Moreover, iReVa is able to perform knowledge withdrawal in almost perfect condition.

Our contributions are summarized as follows: 1) We introduce a novel editing method that initializes and retrofits a key-value adaptor for traceable model editing, which is compatible with most LMs. 2) Our method outperforms recent baselines on model editing tasks with noticeable margins based on various evaluation metrics. 3) We validate the interpretability and generalization capabilities of our method by conducting further analysis such as knowledge withdrawal test and generalization test.

## 2 RELATED WORK

### 2.1 INSIGHT OF KNOWLEDGE STORAGE IN LANGUAGE MODELS

Discussion about how LMs store knowledge has emerged. Petroni et al. (2019) introduced the perspective of treating LMs as knowledge bases and proved its plausibility, which attracted the subsequent attention towards the exploration of the form of knowledge incorporated by LMs. The opinion pointed out by Geva et al. (2021) indicates that factual knowledge is stored in the two-layer-FFN network of a Transformer due to the similar form as key-value memories. This opinion was followed by Li et al. (2024), which further derives the coefficient between final prediction and knowledge neurons in MLP blocks. In contrast, Meng et al. (2023a), through a cosine similarity analysis on hidden states experiment, posed viewpoints that the self-attention module can extract various types of knowledge. Cao et al. (2021) further validates that the weight update is concentrated on parameters in the self-attention module when we train models with new knowledge. Our editing method is built upon the former hypothesis and we focus on the editing to the MLP blocks.

### 2.2 EDITING LMS BY MANIPULATING KNOWLEDGE

With the frequent updates of the knowledge, the demand for model editing increases. Diverse studies have been proposed. By analogy with human knowledge acquisition, we can categorize the editing into three distinct phases. In the recognition phase Bruner (1964), methods such as ERAC and IKE Mitchell et al. (2022a); Zheng et al. (2023) solved the problem by importing additional memories in the form of relevant contexts or prompts. In association phase Bruner (1960), parameter-efficient tuning Hu et al. (2021); Li & Liang (2021); Yu et al. (2023); Hartvigsen et al. (2023) inserts low-rank adaptors or prefix token embeddings to fine-tune new knowledge and combine them to the original models. There are also some studies directly changing the weights of Transformers in the mastery phase Jayashri & Kalaiselvi (2018). For example, Cao et al. (2021) proposed KE, Mitchell et al. (2022a) proposed MEND and Tan et al. (2024) proposed MALMEN to predict the updated parameters of a model with a trained hyper-network. Furthermore, ROME Meng et al. (2023a) and MEMIT Meng et al. (2023b) compute the weight update explicitly with proper representations of knowledge queries and values. However, none of them focuses on traceable model editing, which allows more flexible manipulation of the knowledge.

## 3 PROBLEM FORMULATION

We follow the previous studies Mitchell et al. (2022b); Yu et al. (2023); Hartvigsen et al. (2023) to formulate the task. Suppose we are given a pre-trained language model $f_\Phi$ parameterized by $\Phi$, model editing aims at editing $f_\Phi$ with a dataset $\mathcal{D}_{in} = \{(x_1, y_1), ..., (x_i, y_i)..., (x_n, y_n)\}$, where $(x_i, y_i)$ denotes the edit input-output pairs. Initially, for $x_i \in \mathcal{D}_{in}$, the base model makes prediction $\hat{y}_i = f(x_i)$ but $\hat{y}_i \neq y_i$. In this case, we change $f_\Phi$ by *editing* its parameters to $\Phi^*$. A good model editing to $f_{\Phi^*}$ should satisfy: 1) for any $x_i \in \mathcal{D}_{in}$, the edited model $f_{\Phi^*}$ should output desired predictions, that is $f_{\Phi^*}(x_i) = y_i$; 2) for any input out of the scope of $\mathcal{D}_{in}$, which is denoted as $\mathcal{D}_{out}$, the edited model $f_{\Phi^*}$ should retain the original predictions, that is $f_{\Phi^*}(x_i) = f_\Phi(x_i)$; 3) the edit of $(x_i, y_i)$ towards $f_{\Phi^*}$ should not influence any prior edits $x_{<i} \in \mathcal{D}_{in}$.

## 4 METHOD

To develop an editing method that supports traceable edits to knowledge neurons, we introduce a novel method "**iReVa**" that **i**nitializes and **Re**trofits k**E**y-**V**alue **A**daptors for traceable model editing. The

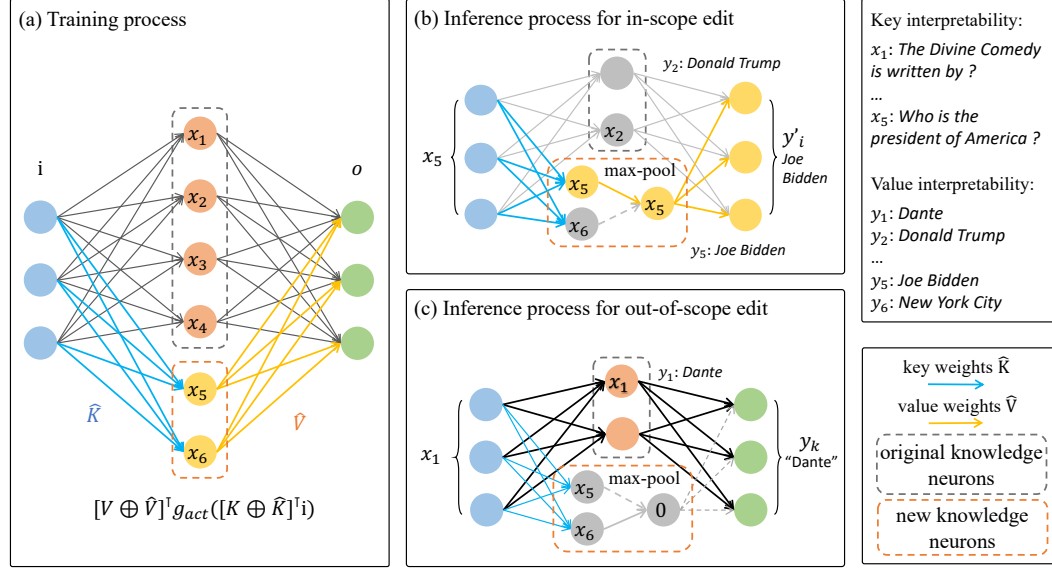

Figure 1: Architecture of iReVa. The left block shows the training procedure with the newly inserted knowledge neurons. The middle block shows the inference procedure with in-scope and out-of-scope edits. We interpret the inference phase by giving some explicit examples (Please note we omit some neurons during inference due to the space limit.). When the query falls in the in-scope edit, our key-value adaptor will be activated and retrieve the corresponding knowledge. When the query falls in the out-of-scope edit, our key-value adaptor is inactive and the model retrieves knowledge from the original memory.

pre-trained LM $f_\Phi$ usually contains Transformer blocks, which consist of intertwined self-attention and feed-forward layers. The prior studies Geva et al. (2021) have shown that the inside MLP blocks are commonly deemed as the neurons for storing implicit knowledge. Our method is able to insert new knowledge without damaging the irrelevant knowledge in the models by inserting and retrofitting the key-value adaptors to these blocks.

Figure 1 depicts the architecture of our proposed method. For a two-layer-FFN MLP block in the $l$-th layer of the original model $f_\Phi$, we denote the weights of the first FFN layer as $\mathbf{K}^l \in \mathbb{R}^{d_1 \times d_2}$ and the second FFN as $\mathbf{V}^l \in \mathbb{R}^{d_2 \times d_1}$. Assume a hidden state $\mathbf{h}^l \in \mathbb{R}^{d_1}$ is an input of the FFN of $l$-th layer, the above block processes the input as follows:

$$\mathbf{i}^l = \text{LAYER\_NORM}(\mathbf{h}^l + \text{SELF\_ATTN}(\mathbf{h}^l)) \tag{1}$$

$$\mathbf{o}^l = \mathbf{V}^{l\mathsf{T}} g_{act}(\mathbf{K}^{l\mathsf{T}} \mathbf{i}^l) \tag{2}$$

$$\mathbf{h}^{l+1} = \text{SELF\_ATTN}(\mathbf{i}^l + \mathbf{o}^l) \tag{3}$$

where $g_{act}$ is the activation layer and $\mathbf{h}^{l+1} \in \mathbb{R}^{d_1}$ is the input of the next Transformer block. Here, $\mathbf{K}^l$ and $\mathbf{V}^l$ emulate neural memories, where keys capture input patterns and values are stored knowledge to be retrieved. When there comes an input vector, it first computes a distribution over the keys, then retrieves the expected knowledge. As the process is just the same for each layer, we can choose any of the layers to edit, we omit $l$ for simplicity in the following description.

Our method inserts a key-value adaptor into the existing MLP block. Specifically, we update $\Phi$ by inserting a new knowledge neuron to store the edit. Two matrices $\hat{\mathbf{K}} \in \mathbb{R}^{d_1 \times n}$ and $\hat{\mathbf{V}} \in \mathbb{R}^{n \times d_1}$ perform as the key-value pair to memorize $n$ edited knowledge, where the knowledge is well-indexed by $n$ dimensions. Therefore, Equation 2 becomes:

$$\mathbf{o} = [\mathbf{V} \oplus \hat{\mathbf{V}}]^\mathsf{T} g_{act}([\mathbf{K} \oplus \hat{\mathbf{K}}]^\mathsf{T} \mathbf{i}) \tag{4}$$

$$= \mathbf{V}^\mathsf{T} g_{act}(\mathbf{K}^\mathsf{T} \mathbf{i}) + \hat{\mathbf{V}}^\mathsf{T} g_{act}(\hat{\mathbf{K}}^\mathsf{T} \mathbf{i}), \tag{5}$$

where $\oplus$ denotes concatenation. As we can see, the key-value adaptor appends more information to $\mathbf{o}$, which could overwrite the original output. And original parameter set $\Phi$ is extended to $\Phi^*$ with the new included parameters $\hat{\mathbf{K}}$ and $\hat{\mathbf{V}}$. Therefore, we aim to find a good key-value adaptor for model

editing that can collaborate with the original knowledge neurons. Considering the independence of the above two function terms and the potential more flexible combination to the output, we relax the formulation of the adaptor to $\text{ADAPTOR}(\mathbf{i}; \hat{\mathbf{K}}, \hat{\mathbf{V}}) = \alpha \hat{\mathbf{V}}^\intercal g_{act}(\hat{\mathbf{K}}^\intercal \mathbf{i})$, which may be a more expressive function with a scaling factor $\alpha$ Hu et al. (2021). Next, we will introduce how to find such an optimal adaptor that not only satisfies the edit success but preserves the original model behavior.

## 4.1 INITIAL KEY-VALUE ADAPTORS FOR IN-SCOPE EDITING

Given an edit $(x_i, y_i) \in \mathcal{D}_{in}$, we first initialize its knowledge neuron $\hat{\mathbf{k}}^0 \in \mathbb{R}^{d_1}$ and $\hat{\mathbf{v}}^0 \in \mathbb{R}^{d_1}$. For $\hat{\mathbf{k}}^0$, we initialize each key to the $x_i$ using the cached input $\mathbf{i}$ predicted by $f_\Phi(x_i)$ at layer $l$, which results in a high probability of matching to the input pattern. For $\hat{\mathbf{v}}^0$, we initialize it using the weights corresponding to $y_i$ from the last layer of $f_\Phi$. Specifically, $f_\Phi(x_i)$ takes charge of generating the next token which can be deemed as the prediction to $x_i$. Thus, we extract the corresponding column of the ground truth token $y_i$ from the weights $\mathbf{W} \in \mathbb{R}^{d_1 \times |V|}$ for generating the next token distribution, where $|V|$ and $d_1$ are the sizes of the vocabulary and dimension of the last layer, respectively [1]. After initialization, we build a mapping from $x_i$ to $y_i$ in a Transformer.

## 4.2 RETROFIT ADAPTORS FOR MODEL EDITING (TRAINING PHASE)

To prevent the effect of the inconsistent scaling brought by built-in parameters in Equation 1, we first normalize $\mathbf{i}$ to ensure that its mean value is close to 0 before it is fed into the adaptor. Given $(x_i, y_i)$, we can have the initialized key-value adaptor as follows:

$$\text{ADAPTOR}(\mathbf{i}; \hat{\mathbf{K}}, \hat{\mathbf{V}}) = \alpha(\hat{\mathbf{v}}^0)^\intercal g_{act}((\hat{\mathbf{k}}^0)^\intercal \mathbf{i}).$$

To avoid the inserted adaptor from distracting the original knowledge stored in existing neurons, we propose to use activation functions that can activate the memory with a large matching value and ignore the memory with a small value. When we deploy the adaptor to models, the activation function usually remains consistent with the base model. Moreover, we apply a hyper-parameter margin $\theta > 0$, which allows memory to be active if $x > \theta$, otherwise inactivate. For example, we use GeLU Shazeer (2020) for GPT Radford et al. (2018) series model and our activation function can be denoted as:

$$g_{act}(x) = \text{GeLU}(x - \theta). \tag{6}$$

The motivations behind the above design in our activation function are two-fold: First, the activation function works as a neuronal inhibitor to inhibit the activation of new knowledge neurons, which retains the original output in most cases. Second, the involvement of the margin further raises the bar to activate the new knowledge neurons. If a certain input is out of the editing scope, it fails to match any memory, all inserted neurons will be inhibited after the activation function as shown in Figure 1.

In practice, edit input $x_i$ is shown in the form of a sequence of tokens such as "{*the*, *capital*, *of*, *China*, *is*}" and $y_i$ is the single-token answer "*Beijing*". This indicates that we have a sequence of hidden states $\{\mathbf{h}_1, \mathbf{h}_2, ..., \mathbf{h}_s\}$ corresponding to input $x_i = \{w_1, w_2, ..., w_s\}$. To avoid damaging the original behavior of the edit model, the edit block merely works on the final token, which is the last token before generation:

$$\text{ADAPTOR}(\mathbf{i}_j; \hat{\mathbf{K}}, \hat{\mathbf{V}}) = \begin{cases} 0 & j \neq s \\ \alpha \hat{\mathbf{V}}^\intercal g_{act}(\hat{\mathbf{K}}^\intercal \mathbf{i}_j) & j = s \end{cases}. \tag{7}$$

where $\mathbf{i}_j$ is the input corresponding to the $j$-th hidden state $\mathbf{h}_j$ in the sequence. As a result, the new knowledge is activated only when the entire input sequence is fed into the model, which not only prevents the dramatic change to the original model but also benefits the gradient update to the key-value pairs[2].

**Fine-tuning adaptors with multiple objectives**. While the above initialization effectively builds the mapping from a certain edit input to the edit output, its impact on irrelevant knowledge may lead to catastrophic forgetting McCloskey & Cohen (1989) issue, which is caused by the extending key-value pairs of the adaptor. In other words, we expect $\text{ADAPTOR}(\mathbf{i}; \hat{\mathbf{K}}, \hat{\mathbf{V}})$ could dominate the

---

[1]See Appendix 9.1 for detailed description of initialization of $\hat{\mathbf{k}}^0$ and $\hat{\mathbf{v}}^0$.

[2]See the discussion of gradient back-propagation of $\hat{\mathbf{k}}$ and $\hat{\mathbf{v}}$ in Appendix 9.2.

output for each $x_i \in \mathcal{D}_{in}$ but maintain unchanged prediction for $x_i \in \mathcal{D}_{out}$ and $x_{<i} \in \mathcal{D}_{in}$. Inspired by the elastic weight consolidation for neural networks Kirkpatrick et al. (2017), we set optimization goals to retrofit $\Phi^*$ with the consideration of the following perspectives.

(1) To maximize the prediction of $y_i$ from the last layer, we maximize the probability of the ground truth edit output given the edit input:

$$\mathcal{L}_{edit} = -\log[\mathbb{P}_{f_\Phi^*}(y_i|x_i)] \tag{8}$$

(2) Even though $\mathcal{L}_{edit}$ enables models to fit the mapping from $x_i$ to $y_i$ effectively, it may push our adaptor far from the initialization, which may damage the initialized key distribution and lead to overfitting. Hence, we propose an additional term to prevent the dramatic change of the update of $\hat{\mathbf{k}}$:

$$\mathcal{L}_{rec} = ||(\hat{\mathbf{k}}^0 - \hat{\mathbf{k}})^\intercal \mathbf{i}||_2^2 \tag{9}$$

(3) Importantly, to prevent the fine-tuning from changing the irrelevant knowledge, we sample some out-of-scope edit data to form $\mathcal{D}_{out}$[3] and retain the original outputs from the model:

$$\mathcal{L}_{irr} = -\frac{1}{|\mathcal{D}_{out}|} \sum_{(x_i, y_i) \in \mathcal{D}_{out}} \max(\hat{\mathbf{k}}^\intercal x_i - \theta, 0) \tag{10}$$

Hence, we comprehend each aspect to form the final objective to retrofit the key-value adaptor:

$$\mathcal{L} = \mathcal{L}_{edit} + a\mathcal{L}_{rec} + b\mathcal{L}_{irr} \tag{11}$$

where $a, b$ are hyper-parameters denoting the importance of the different objective aspects. Note that we edit one knowledge neuron once, but we still support sequential editing by iteratively inserting key-value pairs. During training, all parameters except for $\hat{\mathbf{k}}$ and $\hat{\mathbf{v}}$ for the current edit are frozen. That is, we freeze the prior edit knowledge neurons and simply update the neuron inserted for current edit. This procedure repeats until we have conducted edit over the entire dataset. Compared with parameter high-efficient tuning methods Hu et al. (2021); Liu et al. (2023), which injects the new knowledge into a pre-trained LM as a whole, iReVa focuses on editing parameters in a traceable manner. In other words, we can locate the edited knowledge neurons. At the end, we display the training procedure of iReVa in Algorithm 1.

---

**Algorithm 1** Training Procedure of iReVa

1: **Input** In-scope editing pairs $\mathcal{D}_{in}$; out-of-scope editing pairs $\mathcal{D}_{out}$; Original model $f_\Phi$; Iteration number $T$
2: **Initial** $\Phi^* \leftarrow \Phi$
3: **for** $(x_i, y_i) \in \mathcal{D}_{in}$ **do**
4:     **Initial** $\hat{\mathbf{k}} \leftarrow \mathbf{i}$; $\hat{\mathbf{v}} \leftarrow \mathbf{W}_{[y_i, :]}$     ▷ Initialize key-value adaptor as shown in Section 4.1
5:     $\Phi^* \leftarrow \Phi^* \bigcup \hat{\mathbf{k}} \bigcup \hat{\mathbf{v}}$
6:     **for** $t = \{1, 2, .., T\}$ **do**
7:         $\mathcal{L} \leftarrow \mathcal{L}_{edit} + a\mathcal{L}_{recon} + b\mathcal{L}_{irr}$     ▷ Retrofit key-value adaptor as shown in Section 4.2
8:         $\hat{\mathbf{k}} \leftarrow \text{Adam}(\hat{\mathbf{k}}, \nabla_\mathcal{L}\hat{\mathbf{k}})$
9:         $\hat{\mathbf{v}} \leftarrow \text{Adam}(\hat{\mathbf{v}}, \nabla_\mathcal{L}\hat{\mathbf{v}})$
    **return** $f_{\Phi^*}$

---

### 4.3 ACTIVATE MAX-MATCHING KEY IN ADAPTOR (INFERENCE PHASE)

As we iteratively append $\hat{\mathbf{k}}$ and $\hat{\mathbf{v}}$ to the knowledge neurons. The above procedure will sequentially generate mappings from the edit input to the edit output. Eventually, we obtain two concatenated matrices $\hat{\mathbf{K}} \in \mathbb{R}^{d_1 \times n}$ and $\hat{\mathbf{V}} \in \mathbb{V}^{n \times d_1}$. During inference, we further control the amount of active neurons and highlight the max-matching memory. To this end, we introduce a max-pooling layer to extract the memory with the maximum matching score:

$$\text{ADAPTOR}(\mathbf{i}; \hat{\mathbf{K}}, \hat{\mathbf{V}}) = \alpha \hat{\mathbf{V}}_j^\intercal g_{act}(\hat{\mathbf{K}}_j^\intercal \mathbf{i}), \tag{12}$$

where $j = \text{argmax}_t(\hat{\mathbf{K}}_t^\intercal \mathbf{i})$ and $\hat{\mathbf{K}}_t$ denotes the $j$-th column of $\hat{\mathbf{K}}$. As we can see, when there comes a new input, this layer will highlight the inserted knowledge neurons with the highest similarity to the input as shown in Figure 1. It's worth noting that we exclude the max-pooling layer during the training phase because this may impede the back-propagation due to the inactivation of the neurons.

---

[3]Here, $\mathcal{D}_{out}$ is generated randomly. See Appendix 9.4 for details.

## 5 EXPERIMENTAL SETUP

### 5.1 DATASETS

We perform extensive experiments on two modeling editing tasks: **zsRE** Mitchell et al. (2022a) is a commonly used model editing task derived from a reading comprehension benchmark. Totally $19,086$ examples are included, each example includes a source question, paraphrase question, and corresponding answer. We construct another **PARAREL** Elazar et al. (2021) dataset. Each sentence in PARAREL is derived from a triplet $(s, r, o)$, and the object $o$ was replaced with a "*[MASK]*" token, and a paraphrased version is involved. To apply PARAREL in model editing task, we selected those sentences that end with "*[MASK]*" token to conform to the format of next-token-prediction[4]. For both datasets, we sample irrelevant examples from **NQ** to evaluate the preservation of out-of-scope editing. We test 10K edit in a batch and denote them as **zsRE-**10**K** and **PARAREL-**10**K**, respectively.

### 5.2 BASELINES

We compare our iReVa with 6 advanced baselines that support batch editing: **NO EDITING** denotes we do not modify the base model and utilize its original prediction; **FT** Zhu et al. (2021) is the simple fine-tuning with a constraint on specific parameters. **MEMIT** Meng et al. (2023b) and **ROME** Meng et al. (2023a) are two methods employing a causal analysis to detect the most significant hidden states. They view the editing as a minimum optimization and edit the weight directly, which is effective in batch edit; **MEND** Mitchell et al. (2022a) applies rank-one decomposition to divide the model into two rank-one matrices, which is able to carry mass knowledge in the dense metrics; **MELO** Yu et al. (2023) activates specific LoRA block corresponding to specific queries for multiple edits, which support large-scale editing in just one process. Note that T-Patcher Huang et al. (2023) whose forward propagation resembles our method is not included, now that it can be merely applied on encoder-decoder LMs. Specifically, the patcher is only embedded in the encoder which is inapplicable to the decoder.

### 5.3 EVALUATION METRICS

We follow the commonly-used evaluation metrics Meng et al. (2023a;b) to measure the effect of our editing method.

1. **Edit Success** (ES) measures the models' prediction accuracy on edited data $x_i \in \mathcal{D}_{in}$ by calculating $ES = \frac{1}{N} \sum_{i=0}^{N} \mathbb{I}(y_i = f_\Phi(x_i))$, which represents whether the new knowledge is successfully injected into the base model.
2. **Generalization** (Paraphrase Success, PS) measures the models' prediction accuracy on paraphrase questions provided by benchmarks. We compute paraphrase success with the same formulation but for $x_i$ in the paraphrase questions set. Paraphrase success indicates whether the model can recognize similar expressions and provide edited answers.
3. **Specificity** (Neighborhood Success, NS) measures the models' prediction accuracy on irrelevant questions. Different from $\mathcal{D}_{out}$, these questions are only used for preventing data leakage. We compute neighborhood success with the same formulation but for $x_i$ in the neighborhood questions set. Neighborhood success manifests the capability of solving catastrophic forgetting and preserving irrelevant knowledge stored in model.
4. **Score** is the average of the three aforementioned metrics.

### 5.4 IMPLEMENTATION DETAILS

Regarding editing datasets, we pre-process the edit input-output pairs differently from previous studies. If the multiple tokens form a single prediction, we decompose the multiple tokens into multiple data pairs by greedily appending the previous token in the edit output at the end of the edit input[5]. For model selection, we conduct the experiments on `GPT2-XL` (1.5 Billion parameters) Radford et al. (2019) due to its wide application in existing model editing studies. We trained iReVa on a single NVIDIA A800 80G GPU. On two evaluated benchmarks, we set $a = 1e-3, b = 1e-3, \alpha = 2e-1,$

---

[4]Appendix 9.6 demonstrates the pre-processing step to PARAREL in detail.

[5]The processing procedure is displayed in Appendix 9.5

and iReVa is applied in 47-th (48 layers totally) layer inspired by the assertion in Geva et al. (2021). For the margin in activation function, we set $\theta = 0.75$ for zsRE, $\theta = 0.65$ for PARAREL. During training, we conduct experiments on `GPT2-XL` with setting learning rate as $5e-2$, batch size as 1, and epoch number as 5. We set the learning rate as $5e-3$ for `GPT-J-6B` and apply gradient-free method on `GPT-NEO-2.7B`. More implementation details of baselines are displayed in Appendix 9.7. We re-implement the comparable baselines using the same configuration reported in existing studies.

## 6 RESULTS AND ANALYSES

### 6.1 COMPARISONS TO EXISTING METHODS

Table 1 exemplifies performances of iReVa and baselines on zsRE and PARAREL with 10K edits in batch. As we can see, iReVa outperforms all baselines on average scores with noticeable margins. Even without retrofitting, our method is able to outperform the SOTA results by around $9\%$ and $6\%$ average score improvement on zsRE-10K and PARAREL-10K, respectively. Among all the baseline methods, FT achieves good results on ES and PS, this indicates that fine-tuning is simple but effective to inject knowledge but it could easily distract the irrelevant knowledge, resulting in a poor NS. Whereas other baselines can not guarantee the editing success in a batch, resulting in poor ES and PS. In comparison, iReVa achieves impressive results on all the evaluation metrics. It achieves close to $100\%$ ES without detriment to the original NS. We observe a slight improvement from the results of iReVa to iReVa+$\mathcal{L}$ on zsRE-10K dataset, it verifies our rationale deduce for the initialization of key-value pairs. However, the improvement brought by fine-tuning is not maintained on PARAREL-10K, we suspect this is because the involvement of irrelevant knowledge brings in little unexpected noise with possibility.

Table 1: Editing results on various model editing tasks with `GPT2-XL` as the base model. In our methods, $+\mathcal{L}$ represents iReVa with fine-tuning as described in Section 4.2.

| Method | zsRE-10K | | | | PARAREL-10K | | | |
| --- | --- | --- | --- | --- | --- | --- | --- | --- |
| | Score | ES | PS | NS | Score | ES | PS | NS |
| NO EDITING | 24.17 | 22.89 | 21.96 | 27.65 | 20.03 | 18.66 | 17.24 | 24.18 |
| FT | 57.29 | 82.80 | 64.51 | 24.57 | 52.64 | 83.32 | 53.06 | 21.55 |
| MEND | 15.94 | 12.43 | 12.04 | 23.35 | 0.16 | 0.00 | 0.00 | 0.50 |
| ROME | 11.10 | 17.26 | 14.24 | 1.80 | 5.35 | 9.65 | 6.23 | 0.17 |
| MEMIT | 42.51 | 52.62 | 47.29 | 27.63 | 46.17 | 62.60 | 52.71 | 23.20 |
| MELO | 32.51 | 42.75 | 28.12 | **26.65** | 25.95 | 34.19 | 20.83 | 22.83 |
| iReVa | 66.27 | **97.88** | 74.89 | 26.03 | **58.17** | **93.49** | **56.86** | 24.18 |
| iReVa $+\mathcal{L}$ | **66.77** | 97.47 | **76.38** | 26.47 | 56.80 | 89.85 | 56.37 | **24.18** |

### 6.2 EDIT WITHDRAWAL TEST

Compared with the existing editing methods, our method has the unique advantage of interpretability and traceability, that is we can clearly identify the edit for each newly inserted key-value pair. This provides a chance to conduct an edit withdrawal test. Most existing methods can't perform the withdrawal test for their batch training mechanism, and stream-fashion methods like **GRACE** may encounter the forgetting Hartvigsen et al. (2023) challenge which will induce a withdrawal failure.

Specifically, we test, after editing on 10K examples, if iReVa is able to withdraw certain edits and recover the original output from the base model without much loss. To this end, we inhibit corresponding knowledge neurons as withdrawing the edit, which is denoted as $f_{\Phi^*}^{-\hat{\mathbf{k}}}$. For evaluation, we introduce two metrics, namely **Retrieve Success** and **Consistency**. They are formulated as $RS = \frac{1}{N} \sum_{i=0}^{N} \mathbb{I}(f_{\Phi^*}(x_i) \neq f_{\Phi^*}^{-\hat{\mathbf{k}}_i})$ and $Con = \frac{1}{N} \sum_{i=0}^{N} \mathbb{I}(f_{\Phi}(x_i) = f_{\Phi^*}^{-\hat{\mathbf{k}}_i})$, respectively. The evaluation result on zsRE-10K is shown in Table 2. The results which are close to $100\%$ prove that iReVa can explicitly manipulate the activation of knowledge neurons and easily withdraw the updated knowledge. Notably, this test is not applicable to any other editing methods as their edited parameters are untraceable. This is the first attempt at conducting more flexible knowledge editing.

Table 2: Results of edit withdrawal on zsRE-10K dataset with `GPT2-XL` as the base model.

| Method | Retrieve success | Consistency |
|--------|------------------|-------------|
| iReVa  | 98.02%           | 93.03%      |

## 6.3 EFFICIENCY ANALYSIS

We discuss the spatial and time complexities of iReVa. Regarding time complexity during inference, iReVa only inserts the adaptor in a single $l$-th layer and the insertion only affects the final token prediction of the input. With $\mathbf{i} \in \mathbb{R}^{1 \times d_1}$, $\hat{\mathbf{K}} \in \mathbb{R}^{d_1 \times n}$, $\hat{\mathbf{V}} \in \mathbb{R}^{n \times d_1}$ and averaged length $l$ of target tokens ($l = 2.69$ for zsRE and $l = 1.15$ for PARAREL), the extra time consumption is $\mathcal{O}(ld_1^2 n)$, which is unrelated to the input length and number of layers. Regarding spacial complexity, as we insert two vectors for each edit in a single layer, the extra spacial consumption is $\mathcal{O}(2lnd_1)$. In practice, for `GPT2-XL` with 1.5B parameters, the adaptor merely possesses 0.08B parameters with 10K edits. There is no additional spacial complexity involved in the training phase, given that only $2d_1$ parameters are learnable for each edit token. We empirically record that 10K edits with iReVa cost 7.5/1.6 hours (fine-tuning/without fine-tuning) with a single NVIDIA A800 GPU, compared to 9.16 hours for ROME and 5.4 hours for MEMIT.

## 6.4 ABLATION STUDY

Table 3 shows iReVa's performance on zsRE-10K when we iteratively remove sub-modules: (1) w/o activation function denotes that we remove the activation function proposed in Equation 6. (2) w/o max-pooling denotes that we involve all knowledge neurons during inference instead of the design of Equation 12. (3) w/o $\mathcal{L}_{rec}$ denotes that we train iReVa without initialization and set $a = 0$ in Equation 11. (4) w/o $\mathcal{L}_{irr}$ means we do not apply $\mathcal{L}_{irr}$ by setting $b = 0$ in Equation 11. As we can see, all the modules contribute to the good results. In comparison, the activation function is important to preserve the out-of-scope edit. Without an activation function, we can attain better results on ES and PS, but NS will decrease sharply. We also find that the influence of max-pooling is significant, which may be attributed to noisy data added by a large amount of active but irrelevant knowledge neurons. Besides, excluding $\mathcal{L}_{rec}$ will lead to an observable drop on the three metrics because we discord the effective initialization on $\hat{\mathbf{K}}$ and $\hat{\mathbf{V}}$. Finally, disabling $\mathcal{L}_{irr}$ may induce a marginal improvement in ES and PS, but at the cost of a reduction in NS.

Table 3: Results of ablation study on zsRE dataset with `GPT2-XL` as the base model.

| Activation function | Max pooling | Loss $\mathcal{L}_{rec}$ | Loss $\mathcal{L}_{irr}$ | Score | ES | PS | NS |
|---------------------|-------------|--------------------------|--------------------------|-------|-------|-------|-------|
| ✓ | ✓ | ✓ | ✓ | 66.77 | 97.47 | 76.38 | 26.47 |
| ✓ | ✓ | ✓ | ✗ | 67.00 | 97.84 | 76.73 | 26.43 |
| ✓ | ✓ | ✗ | ✓ | 63.22 | 92.28 | 73.25 | 24.13 |
| ✓ | ✗ | ✓ | ✓ | 44.93 | 56.07 | 52.41 | 26.31 |
| ✗ | ✓ | ✓ | ✓ | 60.27 | 99.41 | 78.52 | 2.87 |

## 6.5 GENERALIZATION CAPABILITIES OF IREVA

**Layer generalization**. To evaluate the effect of iReVa in various layers, we iteratively apply iReVa and the other two baseline editing methods to different layers of `GPT2-XL`, which consists of 48 layers in total. Figure 2 illustrates the influence of three metrics on different layers with intervals. The tendency shows that the edit in the higher layer results in better editing results. This indicates that LMs' final prediction primarily depends on the information retrieved from higher layers and the knowledge stored in lower layers may be overshadowed. For ROME and MEMIT, apparently, they show distinct generalizations in edit layer. Their ES and PS peak at the middle layer like 17 or 22, which proves that the layer generalization is remarkably relevant to the characteristics of different methods. Even though MEMIT achieves good performance in NS when the edit happens in lower layers, overall iReVa outperforms the baselines regarding the comprehensive evaluation metrics.

**LMs generalization**. We also test iReVa on different LLMs as base models, table 4 shows iReVa's generality on different backbones. We apply a larger LM `GPT-NEO-2.7B` Gao et al. (2020),

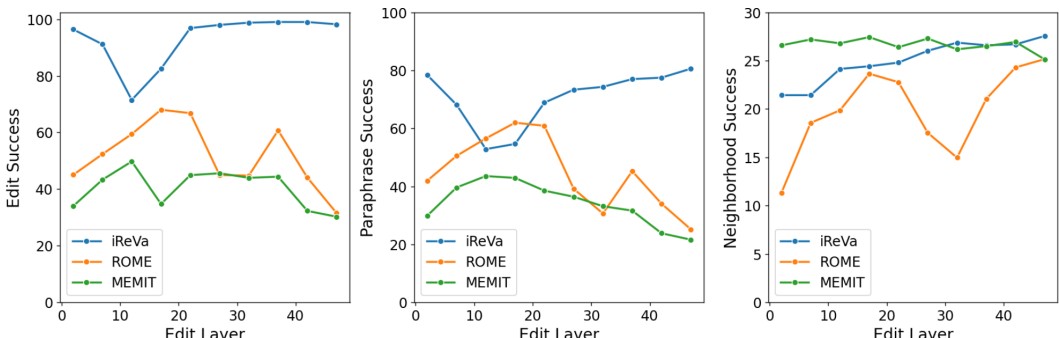

Figure 2: Results of edits in various layers on zsRE dataset with `GPT2-XL` as the base model.

`GPT-J-6B` Wang & Komatsuzaki (2021), and smaller LM `GPT2-LARGE` Radford et al. (2019) to evaluate the effect of iReVa on LMs with different sizes. All of these base models contain two-layer-FFN MLP blocks. IReVa can be deemed as a plug-in module for causal-decoder LMs, which can be applied to more LMs. From the figure, we observe that iReVa can achieve the best average score on all LMs, which shows its general effect.

Table 4: Results on zsRE dataset with `GPT2-LARGE`, `GPT-NEO-2.7B`, `GPT-J-6B` as the base models.

| Engine | Method | Score | ES | PS | NS |
|---|---|---|---|---|---|
| `GPT2-LARGE` | ROME | 29.09 | 38.59 | 36.41 | 12.27 |
| | MEMIT | 43.72 | 56.25 | 49.25 | 25.67 |
| | iReVa | 62.41 | 91.22 | 72.36 | 23.65 |
| `GPT-NEO-2.7B` | ROME | 34.56 | 49.43 | 45.61 | 8.64 |
| | MEMIT | 59.68 | 80.83 | 69.38 | 28.83 |
| | iReVa | 62.20 | 88.23 | 70.71 | 27.66 |
| `GPT-J-6B` | ROME | 40.86 | 53.81 | 49.89 | 18.87 |
| | MEMIT | 66.41 | 94.04 | 72.48 | 32.70 |
| | iReVa | 69.70 | 99.71 | 77.10 | 32.27 |

**Edit quantity generalization**. We discuss the influence on iReVa's performance with the variation of edit quantity, we simply increase the number of edits in the batch and evaluate ES, PS, and NS. Figure 3 shows the tendency of three metrics along with the comparison to baselines ROME and MEMIT. As we can see, iReVa is robust to the number of edit in the batch. It consistently surpasses the other baselines when dealing with the various number of edits. MEMIT performs poorly even with a small number of edits. ROME drops dramatically as the edit number grows.

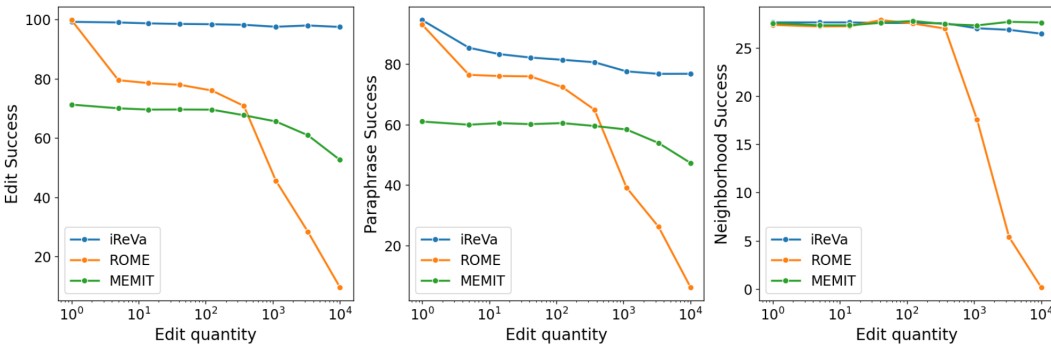

Figure 3: Results of edits with various size on zsRE dataset with `GPT2-XL` as the base model.

## 7 LIMITATION

We also conclude iReVa's limitation as follows: a) iReVa performs poorly when the target prompt is a long sentence because it constructs a knowledge neuron for each token in the target prompt, thereby increasing the training time cost. Additionally, during inference, the high number of neurons increases the probability of errors; b) To maintain iReVa's interpretability, its application is limited, including that iReVa can be only applied on GPT-like models and generation task; c) The behavior of iReVa (ES and PS) won't enhance noticeably as the scale of base model grows.

## 8 CONCLUSIONS

In this paper, we propose iReVa, a model editing method with traceable knowledge storage, which inserts edit key-value adaptor into the MLP module of a transformer model explicitly. iReVa displays prominent abilities of edit success, generalization, and specificity and outperforms baselines with an observable margin. Besides, iReVa first successfully demonstrates its capacity for the knowledge withdrawal. For further research, we will focus on generalizing iReVa to more LM architectures.

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

# 9 APPENDIX

## 9.1 DETAILED DESCRIPTION OF INITIALIZATION OF KEY-VALUE ADAPTOR

We describe how we initialize $\mathbf{k}$ and $\mathbf{v}$ in detail. Given the input $x_i = \{w_1, w_2, ..., w_s\}$, we first obtain the corresponding embeddings for each token, such that $\mathbf{x}_i = \{\mathbf{w}_1, \mathbf{w}_2, ..., \mathbf{w}_s\}$. After encoded via $l$ Transformer layers, we obtain a sequence of hidden representations as input $\{\mathbf{h}_1^l, \mathbf{h}_2^l, ..., \mathbf{h}_s^l\}$. In the two-layer-FFN MLP block of $l$-th layer, after self-attention and layer norm, we have the hidden representation of the last token as:

$$\mathbf{i}_s^l = \text{LAYER\_NORM}(\mathbf{h}_s^l + \text{SELF\_ATTN}(\mathbf{h}_s^l))$$

$$\mathbf{o}_s^l = \mathbf{V}^{l\mathsf{T}} g_{act}(\mathbf{K}^{l\mathsf{T}} \mathbf{i}_s^l)$$

$$\mathbf{h}_s^{l+1} = \text{SELF\_ATTN}(\mathbf{i}_s^l + \mathbf{o}_s^l)$$

We extract $\mathbf{i}_s^{l+1}$ as the initialization of $\hat{\mathbf{k}}^0$. Subsequently, $\{\mathbf{h}_1^{l+1}, \mathbf{h}_2^{l+1}, ..., \mathbf{h}_s^{l+1}\}$ are further processed via the higher layers. In the last layer, we make prediction based on the hidden representation in $L$-th layer, which can be denoted as:

$$P_{f_\Phi}(y_i|x_i) = \text{SOFTMAX}(\mathbf{W}^\mathsf{T} \mathbf{h}_s^L),$$

where $\mathbf{W} \in \mathbb{R}^{d_1 \times |V|}$ and each column denotes the representation of a token. We extract the column corresponding to the ground truth edit out token $y_i$, that is $\hat{\mathbf{v}}^0 = \mathbf{W}_{[:,y_i]}$.

## 9.2 DISCUSSION OF BACK PROPAGATION OF KEY-VALUE ADAPTOR

Recall the knowledge neurons of our key-value adaptor are:

$$\mathbf{o} = \mathbf{v}^\mathsf{T} g_{act}(\mathbf{k}^\mathsf{T}\mathbf{i}) + \hat{\mathbf{v}}^\mathsf{T} g_{act}(\hat{\mathbf{k}}^\mathsf{T}\mathbf{i})$$

Given $\mathcal{L}$, the gradients are computed as:

$$\frac{d\mathcal{L}}{d\hat{\mathbf{k}}} = g_{act}'(\hat{\mathbf{k}}^\mathsf{T}\mathbf{i}) \cdot \hat{\mathbf{v}} \cdot \mathbf{i}^\mathsf{T} \frac{d\mathcal{L}}{d\mathbf{o}}$$

$$\frac{d\mathcal{L}}{d\hat{\mathbf{v}}} = g_{act}(\hat{\mathbf{k}}^\mathsf{T}\mathbf{i}) \frac{d\mathcal{L}}{d\mathbf{o}}$$

$$\frac{d\mathcal{L}}{d\mathbf{i}} = [g_{act}'(\mathbf{k}^\mathsf{T}\mathbf{i})\mathbf{v}^\mathsf{T}\mathbf{k} + g_{act}'(\hat{\mathbf{k}}^\mathsf{T}\mathbf{i})\hat{\mathbf{v}}^\mathsf{T}\hat{\mathbf{k}}] \frac{d\mathcal{L}}{d\mathbf{o}}.$$

where $g_{act}'$ is the derivative of the activation function. We have multiple observations of the gradients: First, we would like the newly inserted neuron to be activated initially, namely $g_{act} > 0$. Otherwise, the gradients are close to $0$ and the neurons are likely to be dead. This is the reason why we initialize the $\hat{\mathbf{k}}$ and $\hat{\mathbf{v}}$ with the consideration of having a high matching value. Second, when we update $\hat{\mathbf{k}}$ and $\hat{\mathbf{v}}$, they are unrelated to $\mathbf{k}$ and $\mathbf{v}$, which makes it possible to isolate the irrelevant knowledge.

For the knowledge neurons without our key-value adaptor, we have the propagation:

$$\mathbf{o} = \mathbf{v}^\mathsf{T} g_{act}(\mathbf{k}^\mathsf{T}\mathbf{i}).$$

The gradients of $\mathbf{i}$ are computed as:

$$\frac{d\mathcal{L}}{d\mathbf{i}} = g_{act}'(\mathbf{k}^\mathsf{T}\mathbf{i})\mathbf{v}^\mathsf{T}\mathbf{k} \frac{d\mathcal{L}}{d\mathbf{o}}.$$

As we can see, excluding the key-value adaptor in the neuron makes the gradients simply derived from $\mathbf{k}$ and $\mathbf{v}$, which maintains the original knowledge in the neurons.

## 9.3 Influence of $\theta$ and $a$

The influence of $\theta$ is illustrated in 9.3. The figure shows the trade-off between the three metrics smoothly. The primary affected metric is **Neighborhood Success**, and **Edit Success** and **Paraphrased Success** exhibit a slight downward trend. For $a$, we find that merely **Paraphrase Success** peaks while $a = 1e - 2$, meanwhile **Edit Success** and **Neighborhood Success** do not continue to improve with the increase of $a$.

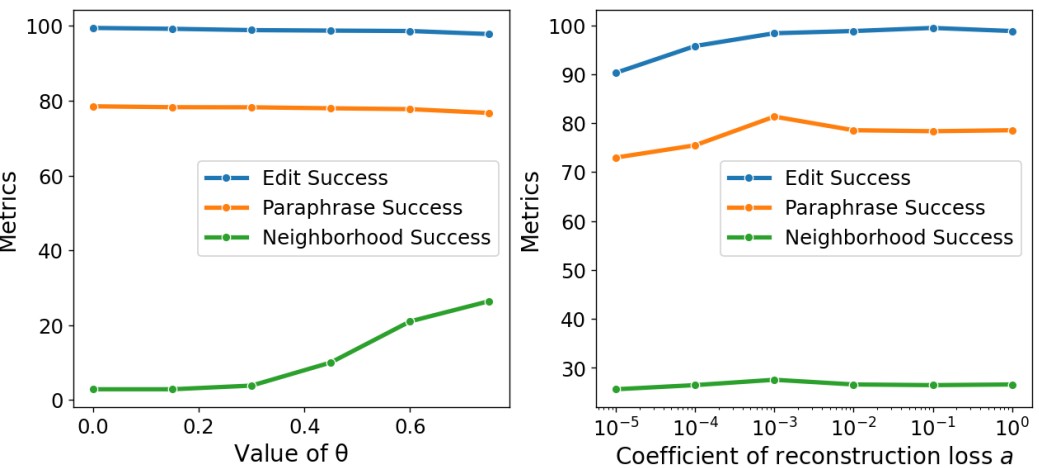

Figure 4: Correlation between three metrics and $\theta$(left) or $a$(right) of iReVa, ROME, MEMIT

## 9.4 Sample out-of-scope examples for iReVa

To enhance iReVa's Specificity, we generate 3 kinds of irrelevant questions $q \in \mathcal{D}_{out}$ for each $(x, y) \in \mathcal{D}_{in}$ to minimize $\hat{\mathbf{K}}_i^{\mathsf{T}} \cdot x_{out}$, where $x_{out}$ is the dense representations of $q$. These questions are listed as follows: a) Randomly generated questions produced by feeding base model with a `bos` (begin of sentence) token. b) Questions generated by base model with feeding the subject $s$ of the $x$ provided by the benchmark. c) Questions sampled from other examples in training dataset, whose opinion is similar to contrastive learning Hadsell et al. (2006). During iReVa training, we generate 2 questions in a), 2 questions in b), and 6 questions in c) for each training example.

## 9.5 Pre-processing procedure of zsRE

Shown in 2, we split each $(x, y)$ pair into multiple $(x', y')$ to ensure $y'$ is a single-token edit out. This procedure is also applied in the evaluation of zsRE and PARAREL, which measures the $(i + 1)$-th token of edit-out prediction accuracy given edit-in and $i$ prefixes of edit-out.

---

**Algorithm 2** Pre-processing Procedure of PARAREL

1: **Input** Raw dataset zsRE $\mathcal{D}$, tokenization function $\mathrm{encode}$;
2: **Init** $\mathcal{D}' = []$;
3: **for** $(x, y) \in \mathcal{D}$ **do**
4:     **Init** tokens $= \mathrm{encode}(y)$;
5:     **for** $i \in \{0, 1, 2...\mathrm{len}(\mathrm{tokens}) - 1\}$ **do**
6:         $\mathcal{D}'.append((x + \mathrm{tokens}[: i], y[i]))$;
    **return** $\mathcal{D}'$

---

## 9.6 Pre-processing Procedure of PARAREL

This section details the pre-processing method on close text dataset PARAREL Elazar et al. (2021). PARAREL contains 34 types of relations $r$, with an average of 900 question bags $b$ per relation,

---

**Algorithm 3** Pre-processing Procedure of PARAREL

---

1: **Input** Raw dataset PARAREL $\mathcal{D}$; Raw NQ dataset $\mathcal{D}_{loc}$; Function lcs computes the longest common sub-array of two strings, tokenization function encode, detokenization function decode;
2: **Init** $\mathcal{D}' = []$;
3: **for** $(r_i, v_i) \in \mathcal{D}$ **do**                 ▷ For each relation and in-relation questions in $\mathcal{D}$
4:     **for** $(b_{ij}, a_{ij}) \in v_i$ **do**      ▷ For specific questions, rephrased versions and answers in $v_i$
5:         **If** $\text{len}(b_{ij}) \leq 1$, **then continue**;
6:         **Init** subject $= b_{ij}[0]$;
7:         **Init** compatible_questions $= []$;
8:         **for** $q_{ijk} \in b_{ij}[1:]$ **do**
9:             subject $= \text{lcs}(\text{encode}(q_{ijk}), \text{encode}(\text{subject}))$;
10:             **If** $q_{ijk}.endswith("[MASK]")$, **then** compatible_questions$.append(q_{ijk})$;
11:         src_question $=$ compatible_questions$[0]$;
12:         subject $= \text{decode}(\text{subject})$
13:         **If** (subject $= ""$) $\vee$ (subject $=$ src_question), **then continue**
14:         rephrased_question $= random.choice(\text{compatible\_questions}[1:])$;
15:         $\mathcal{D}'.append((\text{src\_question}, a_{ij}, \text{rephrased\_question}, \text{subjcet}, \mathcal{D}_{loc}.next()))$
16: **return** $\mathcal{D}'$

---

totaling 27,738 distinct questions $q$. And for each question bag, around 9 rephrased versions are recorded with a sole answer $a$.

The entire pre-process algorithm is shown in 3. To make PARAREL applicable for the next-token-prediction task, we reserve the sentences that end with a special token "*[MASK]*". After a round of filtering, we removed question bags $b$ with only 1 valid sentence that ends with "*[MASK]*" for both **Edit Success** and **Paraphrase Success** need to be computed. During this filtering, we collect the subject of question $s$ bag by calculating the longest common sub-array of all $q \in b$ tokenized by `GPT2Tokenizer` Radford et al. (2019) simultaneously for specific methods require the subject of a question. The next screening occurs at $b$ whose subject $s$ is an empty string or identical to $b[0]$. With residual question bags $b'$, we choose $b'[0]$ as the source question and a randomly sampled question from $b'[1:]$ as the paraphrase question.

Empirically, we believe PARAREL is harder than zsRE because the average token length of edit target is shorter, thus model can't give more empirical predictions based on given prefix of the target, which is mentioned in 9.5. In other words, the account for first-token prediction may influence the difficulty of datasets noticeably.

## 9.7 IMPLEMENTATION DETAILS OF COMPARABLE BASELINES

### 9.7.1 FINE TUNING(FT)

We implement fine tuning on two feed-forward networks (`mlp.c_fc, mlp.c_proj`) at the layer of 46 with `GPT2-XL`. The base model is trained for 20 epochs with $lr = 1e-4, \text{batch size} = 32$.

### 9.7.2 MEND

We do not load the pre-trained MEND Mitchell et al. (2022a) weight, but apply MEND directly. Hyper-parameters of MEND keep consistent with the configuration of MEND's open-source code.

### 9.7.3 ROME, MEMIT

ROME Meng et al. (2023a) and MEMIT Meng et al. (2023b)'s setups on **GPT2-XL** also remain identical to the source code. On `GPT-NEO-2.7B` and `GPT-J-6B`, we alter the edit layer to 5 for ROME and {3,4,5,6,7,8} for MEMIT.

### 9.7.4 MELO

Due to larger edit amount and different backbone for zsRE, we modify several configurations to make MELO Yu et al. (2023) comparable to our methods. For MELO's code book, we enlarge the number of blocks (clusters) to 100. Besides, we rewrite MELO's training loss to make it compatible with causal decoder.

