# OpenReview forum: "Initializing and Retrofitting Key-Value Adaptors for Traceable Model Editing"
_ICLR.cc/2025/Conference — Submitted to ICLR 2025_

### Official Review · Reviewer_bY48 · 2024-10-31

**Soundness:** 2
**Presentation:** 3
**Contribution:** 2
**Rating:** 3
**Confidence:** 4

**Summary:**

This paper proposes a new method to perform model-editing and allow potential knowledge withdrawing.

**Strengths:**

- The topic is important.
- The proposed idea is simple.
- The experimental results seem to show the effectiveness of this method

**Weaknesses:**

- **Method Design**: The paper proposes to add the adaptor to the original model, however, as stated in line 201 "To avoid damaging the
original behavior of the edit model, the edit block merely works on the final token, which is the last token before generation", this means some **oracle information** is used in this model, i.e., **this method needs to let the model know which is the final token**. This is impractical in real world. When we edit the knowledge in the model, we want the model to answer correctly no matter what users ask, and we would never know when the model is going to reveal the knowledge that is supposed to be edited. For instance, when the knowledge "sky is blue" is edited to "sky is green", then for various questions such as "is the color of the sea and the sky the same?" the model would fail as this method would not know when to add the adaptors.

- **Experimental Results**: For zsRE-10k, the authors did not use the deduplicated dataest from MEMIT, which may yield unfair comparisons. As the results in the original paper show that MEMIT can achieve 96.7 (ES), 89.7 (PS) and 26.6 (Specificity) on 10000 edits, and it is only 52.62, 47.29, 27.63 as reported in this paper. I would doubt if the implementation is correct and the hyper-parameter is properly tuned. The ideal case would be evaluating iReVa on the exactly same dataset used in MEMIT.

- **Withdrawing knowledge experiments**: The authors stated in line 377: "Notably, this test is not applicable to any other editing methods as their edited parameters are untraceable. This is the first attempt at conducting more flexible knowledge editing." However, It is feasible to withdraw knowledge from MEMIT, GRACE, etc. Please refer to [2], where the authors withdraw the knowledge by editing "The president of United States is Joe Biden" to "The president of United States is <endoftext>", i.e., using the token "<endoftext>" can allow these model-editing methods to edit the model, which shows pretty good results. Besides, it seems to be quite trivial for this method to be able to withdraw the knowledge as they can just remove the related adaptors.

[1] Mass-Editing Memory in a Transformer
[2] Large Scale Knowledge Washing

**Questions:**

I do not have extra questions.

---

> ### Author Response · Authors · 2024-11-19
>
> ### Reply to Reviewer bY48
>
> Thank you, Reviewer bY48, for your valuable feedback! We will address the weaknesses you highlighted in this reply.
>
> #### Weakness 1 (Method Design):
> You mentioned that some oracle information is used, as the model does not know the final token. Actually, the final token refers to the last non-padding token in the input prompt, which the model is aware of. Regarding your example ("sky is blue" → "sky is green"), when asked, "Is the color of the sea and the sky the same?", this involves the Portability metric, which evaluates indirect reasoning problems related to the edited knowledge. Current methods, including ours, focus on direct editing to answer specific questions.
>
> You suggested activating the adaptor at an unknown intermediate token for reasoning tasks. While this is a promising idea, the reasoning capabilities of language models remain poorly understood, thus we did not attempt this mechanism in our approach. One potential solution is to use Chain-of-Thought (CoT) [1] prompting, allowing the model to generate reasoning step-by-step. For more details on Portability, please refer to Section 3 of the Author Global Rebuttal.
>
> #### Weakness 2 (Experimental Results):
> We have addressed this in Section 1 of the Author Global Rebuttal. If further clarification is needed, please let us know.
>
> #### Weakness 3 (Withdrawing knowledge experiments):
> The Withdrawal test aims to evaluate the flexibility of managing applied edits, particularly for knowledge that requires frequent updates. This does not assess the ability to delete existing knowledge directly. In the Withdrawal test, we independently retract a knowledge edit and observe if the model reverts to its pre-edit state. MEMIT [2] writes update matrices for a batch of edits, making it inflexible to retrieve one edit as the entire batch must be reverted. GRACE [3] cannot complete this test, as noted in L368-L369. You mentioned that iReVa can achieve this by simply removing related adaptors, and this indeed highlights iReVa's advantage over batch-edit methods like MEMIT.
>
> #### Reference
>
> [1] Chain-of-thought prompting elicits reasoning in large language models
>
> [2] Mass-Editing Memory in a Transformer
>
> [3] Aging with grace: Lifelong model editing with discrete key-value adaptors

---

> > ### Comment · Reviewer_bY48 · 2024-11-22
> > **Response to the Rebuttal**
> >
> > Thanks for addressing my concerns.
> >
> > [W1] The authors mention that `Regarding your example ("sky is blue" → "sky is green"), when asked, "Is the color of the sea and the sky the same?", this involves the Portability metric, which evaluates indirect reasoning problems related to the edited knowledge. Current methods, including ours, focus on direct editing to answer specific questions.` However, current methods do not only focus on direct editing to answer specific questions, that's why there are metrics `Efficacy` and `Generalization` in MEMIT.  Although during their evaluation, they still use prompts to evaluate `Generalization` there is no additional operation at the end of the step, which can make people believe that the model's behavior is not dependent on the prompt, rather, the knowledge has been washed. However, in your method, only when the model is using the adapter when generating every token can it make people believe that it will have similar behavior and similar practicality as MEMIT.
> >
> > [W2] **as the model can answer some zsRE questions correctly by existing knowledge obtained in pre-training phase** This doesn't make sense. GPT2-XL can only answer around 10% of zsRE questions. I do believe that the code is correct, but not using the same setting as in MEMIT makes me feel insecure about the results. It also doesn't make sense to discard the setting as in MEMIT just because this is knowledge-insertion rather than knowledge-editing because your method do not require the model to acquire the original knowledge before editing, you are still doing insertion. (As stated in line 152 `Our method inserts a key-value adaptor into the existing MLP block.`)
> >
> > [W3] For MEMIT, they can simply run ROME to delete knowledge. Although I do acknowledge that they fail at sequential editing so this way is probably not gonna work well. I believe iReVa has the advantages over previous methods when it comes to retreating knowledge. This point can be seen as resolved.
> >
> > [W1] and [W2] were my major concerns and they are still here. I would love to keep my rating.

---

> > > ### Author Response · Authors · 2024-11-22
> > >
> > > # Reply to Reviewer bY48
> > >
> > > Thank you, reviewer bY48, for your valuable feedback! In this reply, we aim to address the questions and concerns you raised in your previous comments.
> > >
> > > ---
> > >
> > > - **W1:** We understand your concern regarding the reliability of iReVa's explicit addition of adaptors compared to the implicit knowledge update mechanisms used in methods like MEMIT. This distinction indeed highlights a key difference between the two approaches, but it does not necessarily imply that one is inherently superior to the other. Both methods have their strengths and limitations.
> > >
> > >   Our approach is admittedly more suited for direct editing tasks, which can be considered a limitation. However, MEMIT encounters challenges when the training targets conflict with the model's pre-existing knowledge. The discrepancies in MEMIT's performance on `zsRE (target_true)` (as reported in the MEMIT paper) and `zsRE (target_new)` (as shown in our experiments) highlight this issue. MEMIT lacks robustness to training targets, which can reduce its reliability and interpretability.
> > >
> > >   In contrast, iReVa is highly interpretable. By employing an insertion-based approach, it can effectively overwrite conflicting knowledge, ensuring the model does not struggle to reconcile old and new information.
> > >
> > >   Although MEMIT's results on CounterFact demonstrate its ability to increase the probability of `target_new` over `target_true`, its performance on zsRE indicates that it cannot directly output `target_new` as the final prediction. This limitation reduces its practical utility, as one of the key goals of knowledge editing is to enable the model to provide updated answers.
> > >
> > >   Another important application of knowledge editing is **portability (ripple effect)**. While MEMIT's implicit knowledge update mechanism seems advantageous in this regard, the lack of interpretability in language models presents a significant challenge. As a result, methods like MEMIT, iReVa, and many others struggle to optimize for portability. In fact, prompting-based methods currently exhibit better performance in this area.
> > >
> > >
> > > - **W2:** You pointed out that "`GPT2-XL can only answer around 10% of zsRE questions`" does not mean that GPT2-XL only contains 10% of the knowledge in the zsRE dataset. Some knowledge may exist in the model but cannot be directly predicted. Thus, the actual proportion could exceed 10%. Additionally, the results reported in the MEMIT paper are based on GPT-J-6B, and the differences between GPT-J-6B's performance in the MEMIT paper and our experiments might align with this proportion. For instance, even if GPT-J-6B only knows 20% of the knowledge in zsRE, the observed difference in paraphrase accuracy metric (89.7 in MEMIT vs. 72.48 in our paper) could correspond to this knowledge disparity.
> > >
> > >   You also mentioned that since our method is insertion-based, there is no need to differentiate between knowledge insertion and knowledge update. However, these two tasks fundamentally differ based on whether the training knowledge conflicts with the model's pre-existing knowledge (as noted in W1). The supplementary experiments on MEMIT with LlAMA-3, included in our Global Rebuttal, further demonstrate this issue. Since LlAMA-3 likely knows a larger proportion of zsRE knowledge than GPT-J-6B, it further highlights this distinction. In practice, our method is unaffected by this 20% knowledge gap, which instead highlights its robustness and advantages.

---

> > > > ### Comment · Reviewer_bY48 · 2024-11-23
> > > > **Response to the authors**
> > > >
> > > > **[W1]**: (1) If the method can only work under **direct editing** setting, it would be too limited to be used.  (2) Then why don't the authors compare with baselines on CounterFactual? As said, this dataset is more suited to the paper's setting as well.
> > > >
> > > > **[W2]**: The authors are basically saying the base model knows a lot about the knowledge in zsRE (although GPT2-XL might only knows under 20% of the knowledge) so this setting cannot be adopted. I don't understand this, if the model already knows the knowledge, then why would the current method fail? Intuitively it would strengthen the master of existing knowledge and edit the model with its unknown knowledge.
> > > >
> > > > In summary, the paper has some severe drawbacks (as said in [W1] and acknowledged by the authors); The experiments are kind of strange as the authors modified zsRE setting and skipped CounterFactual, which is hard to convince people the effectiveness of this method.

---

> ### Comment · Reviewer_bY48 · 2024-11-23
> **Additional Comments**
>
> I just looked at the authors' reponse to other reviewers and saw the clarifications on CounterFactual. However, I don't agree with the authors' statement `Regarding CounterFact, we believe its purpose is not to test the prediction accuracy of editing methods but to evaluate whether the model can implicitly increase the probability of the new target. The datasets we use, however, aim to test the model's ability to explicitly predict the new target. Therefore, we did not use CounterFact.` I don't think `CounterFact is to evaluate whether the model can implicitly increase the probability of the new target`. That was simply the metric in MEMIT but not the goal of this dataset. In this dataset, every instance has `target_new` which can be simply used in the paper's setting.

---

### Official Review · Reviewer_WcNm · 2024-11-02

**Soundness:** 3
**Presentation:** 3
**Contribution:** 2
**Rating:** 6
**Confidence:** 4

**Summary:**

This work proposes an alternative methodm, **iReVa**, to update/insert $(s, r, o)$ knowledge tuples in autoregressive transformer LMs. The authors propose expanding number of neurons in the middle representation (output of `up_proj`) of 2 layer MLP blocks (`up_proj` followed by a `down_proj`). A (set of) such additional neurons uniquely correspond to an updated knowledge tuple, and the authors showed that they can leverage this to "turn off" those neurons to retrieve the LMs original prediction before that specific update.

**Strengths:**

* The proposed method scales to batch updates upto 10K knowledge tuples. Many of the knowledge editing methods fail to reach that kind of scale.
* The added neurons are traceable to specific knowledge tuples, and thus the method is more interpretable.
    * For frequently changing facts a set of iReVa neurons can also be reusable (I think, the authors didn't really mention this in the paper)
* Seems to beat other methods in benchmarks (I am a bit suspicious on this, see questions)

**Weaknesses:**

I think the paper introduces a nice idea and is decently written. But I have some concerns about the scores presented in their evaluations as they don't match the scores reported in existing works (see Questions, please). On that ground I am choosing a borderline reject. I will be happy to increase my score if the authors can give reasonable explanations.

**Edit:** Score increased to 6 (borderline accept) after authors' rebuttal.

**Questions:**

* You mentioned CRUD operations in the abstract. Do you think your method can be applied to **delete** an existing knowledge from the LM?

* Do you add one neuron (one K row and one V column) per knowledge tuple? Or, is that $n$? I am assuming $n$ is the batch size (?), but I got a bit confused later by some of the languages in the paper.

* For multi-token objects such as $P(s, r) = $ `Ran Blake used to teach in` $o = $ `New England Conservatory of Music`, you split that tuple into multiple individual facts. You split it into $P(s, r) \rightarrow$ `New`, $P(s, r) +$ `New` $\rightarrow$ `England` ..., if I am not wrong. I feel you need further tests to make sure if that is alright. Does the model forget to map `New` to other valid continuations, like `New Zealand`? I think this should be tested with targetted cases designed specificly for the multi-token object in question. Randomly sampling unrelated facts and doing a specificity test is not enough to address this issue.

* Evaluation

    * The score $S$ used in ROME, MEMIT ([Meng et al, 2022](https://arxiv.org/pdf/2202.05262)) is the ***harmonic*** mean of $ES$, $PS$, and $NS$; which penalizes more for lower individual scores. Fix this in your paper.

    * You didn't use CounterFact (by [Meng et al, 2022](https://arxiv.org/pdf/2202.05262)) or other datasets, but proceeded to make your own dataset. I am not sure what prompted you to do this considering that your dataset is very similar, both in structure and in scale, to CounterFact (I think). And if I know correctly CounterFact also adapts zsRE, PARAREL, (and WikiData). Did you find some limitations in the existing dataset/benchmarks?

    * Can you give examples of what kind of paraphrases you test generalization (PS) with? I am a bit surprised that you were able to reach this good generalization scores by targeting the last token of the input pormpt $P(s, r)$. If I understand right, the main reason ROME/MEMIT targets subject last position instead is to achieve better generalization. You should also check cosine similarity of representations with different paraphrases to justify this design choice.

    * I was surprised to see such poor scores for MEMIT on Table 1. I expected atleast the efficacy score (ES) to remain high across all the LMs as MEMIT calculates $V$ of the $K \rightarrow V$ map with a gradient optimization. MEMIT reported scores on GPT-j (Figure 5 on CounterFact 10K and Table 1 on zsRE 10K,  [Meng et al, 2023](https://arxiv.org/pdf/2210.07229)), and you also include GPT-J results on Table 4. Your scores just doesn't seem to match. This makes me suspicious of your reported scores. (I am choosing to believe Meng et al's reported scores over yours as their paper is already published and multiple followup works has evaluated and extended that work.) Are you applying MEMIT on the last token of the prompt instead of the last token of the subject? Is it possible that you have made some other errors while setting up these benchmark methods? ... As far as I could understand, your dataset is not that different to justify this discrepancy.

---

> ### Author Response · Authors · 2024-11-19
>
> ### Reply to Reviewer WcNm
>
> Thank you, Reviewer WcNm, for your valuable feedback! We will address the questions you raised in this reply.
>
> #### Question 1 (Knowledge Delete):
> Thank you for pointing this out. iReVa can indeed delete existing knowledge. Specifically:
> - To prevent the model from answering a question, we initialize the value vector $v$ in the $(k, v)$ pair with the embedding of the |eos| token ($W_{eos}$).
> - To reduce the likelihood of predicting a specific answer $a$ for a question $q$, we can initialize $v$ with $-W_a$, effectively lowering $a$'s prediction probability.
>
> #### Question 2 (Adding one neuron per knowledge tuple):
> Apologies for not making this clear in the paper. We insert one column in $K$ and one row in $V$ for each knowledge tuple. For $n$ knowledge tuples, $n$ columns and rows are added, followed by unified testing. During training, the batch size is a hyperparameter, and we set $batch\_size = 1$ to ensure independence between multiple knowledge samples.
>
> #### Question 3 (Multi-token objects):
> You raised a concern about the model being influenced by the last token of multi-token objects, leading to predictions directly related to that token. If iReVa adaptors are applied in lower layers of the model, this issue could occur due to insufficient context. However, iReVa operates on the penultimate layer, where the last token has already incorporated sufficient context information, avoiding reliance solely on itself. The high ES metric in our paper's main table supports this claim.
>
> #### Question 4 (Evaluation):
> ##### Question 4.1 (Harmonic mean):
> Thank you for pointing out the error. We will correct this in the revised version, and the new $S$ metric will not affect the comparison between iReVa and other methods.
>
> ##### Question 4.2 (CounterFact):
> We did not evaluate iReVa on the CounterFact dataset from ROME [1] due to its high ambiguity. Metrics for CounterFact is a probability-comparison based metrics, and they only require the new answer's probability to surpass the original answer's probability. In MEMIT, all training memories are integrated into the model simultaneously, enabling the new answer's probability to exceed the original for all samples. In contrast, iReVa trains each example independently, activating only one example's memory during testing. iReVa is essentially unable to achieve a score on these metrics when incorrect memory is activated, and the ambiguity of the test cases in this dataset significantly impacts the performance of our method. Our results on CounterFact are as follows:
>
> |  Backbone   | Method  | S $\uparrow$ | ES $\uparrow$ | PS $\uparrow$ | NS $\uparrow$ | GE $\uparrow$ | RS $\uparrow$ |
> |-------------|---------|:-----:|:-----:|:-----:|:-----:|:-----:|:-----:|
> | GPT-J-6B    | MEMIT   | 85.40 | 98.75 | 87.45 | 73.70 | 618.5 | 39.84 |
> |             | iReVa   | 61.20 | 99.53 | 42.83 | 64.00 | 621.0 | 30.90 |
>
> ##### Question 4.3 (Paraphrase examples):
> We apply the adaptor to the last token because autoregressive models use its representation in the final layer to predict the next token. During editing phase, we will initialize $k_i$ with the representation of original question and $v_i$ with corresponding target. For paraphrase examples, the correct $k_i$ and $v_i$ are activated if the dot product between the paraphrase embedding $x$ and edit question representations $k_i$ exceeds that with any other $k_j (i \neq j)$. This depends on the model's ability to encode similar representations for paraphrase and original questions, and experiment results demonstrate this ability of our chosen backbone. For instance:
> - Source question: "Who is the architect for Toodyay Fire Station?"
> - Paraphrase: "Who was responsible for the planning of the Toodyay Fire Station?"
> The cosine similarity is 0.9107, far exceeding the second-highest score of 0.6187.
>
> ##### Question 4.4 (Comparison with MEMIT):
> We have addressed this in Section 1 of the Author Global Rebuttal. If further clarification is needed, please let us know.
>
> #### Reference
>
> [1] Locating and Editing Factual Associations in GPT

---

> > ### Comment · Reviewer_WcNm · 2024-11-19
> >
> > I thank the authors for their response. Please find my responses to your rebuttal below:
> >
> > ### **Global Rebuttal**
> > > *"... LlAMA models have an architecture incompatible with some baseline methods like MEMIT."*
> >
> > It is true that the Llama models (and most standard transformer LMs) use gated MLP instead of a standard 2 layer MLP. But, MEMIT works on the *down* projection, $D$ is authors' notation. The $k$ and $m$ in the MEMIT paper are the inputs and outputs of a standard down projection, which is the same for both gaged MLP and standard 2 layer MLP. So, I don't see why MEMIT should not work on Llama models. Can you please elaborate if I am missing something?
> >
> > > *"... whereas MEMIT uses the ground-truth answer, as evidenced by their source code (/dsets/zsre.py)"*
> >
> > zsRE is question-answering task. MEMIT and other subsequent works tests how such methods can add *correct* knowledge to the LMs. But thanks for the clarification that you use the alternative answer for this work.
> >
> > However, later you say that in your PARAREL dataset you use the ground truth answer. I think this is a bit confusing.
> >
> > ### **Rebuttal for my questions**
> > * Question 1 (Knowledge Delete):
> >     * If we map $k$ to a $v$ that is the `<|eos|>` token (or all zeros), will this be a language model anymore? The LM still needs to be fluent in the language, right?
> >     * The second approach is more like updating with another $v$ that is the negation of the original $v$. I think this is a bit more reasonable, but needs to be tested if it works.
> >
> >     This was a bit of a far-fetched question anyways. I appreciate your response.
> >
> > * Question 2 (Adding one neuron per knowledge tuple): Thanks for the clarification. I suggest you make this clear in the paper.
> >
> > * Question 3 (Multi-token objects): I don't see how the higher ES metric in Table 1 is supposed to support your claim about LMs developing a good understanding of the context. Don't you always measure ES with the prompt that was used to extract $k$? Did you mean PS here by any chance? Requesting further clarification.
> >
> >     And, overall, I find the answer unconvincing and I still think you need to test this with more targeted cases.
> >
> >
> > * Question 4 (Evaluation):
> >
> >     * Question 4.2 (CounterFact): Thanks for the clarification. I think it is an important distinction that you are using a harder metric about the top prediction instead of comparing probabilities. But still I fail to understand how your dataset is structually different. You could have used CounterFact and just changed the evaluation metric, right? Thanks for providing the scores. But I am very confused how these two very similar datasets seem to favor two different methods by this large of a margin.
> >
> >     * Question 4.3 (Paraphrase examples): Thanks for the clarification. I still think this approach is problematic as the answer $v$ will be very tightly bound to the question $P(s, r)$. I mean the edit will simply fail to generalize if you ask a follow-up question like `What is the nationality of the architect who designed Toodyay Fire Station?`. But it seems like ROME/MEMIT is also not very good at these ripple effects of knowledge editing ([Cohen et al, 2024](https://aclanthology.org/2024.tacl-1.16.pdf)). But it is another important metric to test.

---

> > > ### Author Response · Authors · 2024-11-20
> > >
> > > # Reply to Reviewer WcNm
> > >
> > > Thank you, Reviewer WcNm, for your response! In this reply, we will address the questions raised in your previous comments.
> > >
> > >
> > > ## Global Rebuttal
> > >
> > > ### On the LlAMA family models and MEMIT
> > >
> > > You are correct that MEMIT operates on the second MLP within the two-layer FFN (referred to as $K$ and $V$ later). Specifically, it modifies the $V$ matrix. Even within the LlAMA series models, the corresponding Down matrix $D$ can also be regarded as serving the same function as $V$. We have also added experiments on LlAMA3, with results as follows. Considering the time overhead, we only tested 1K examples.
> > > |  Backbone   |    Method     | S $\uparrow$  | ES $\uparrow$ | PS $\uparrow$ | NS $\uparrow$ |
> > > | :---------: | :-----------: | :---: | :---: | :---: | :---: |
> > > |             | NO EDITING(1K)| 35.99 | 32.36 | 31.12 | 49.19 |
> > > | LlAMA3.1-8B |   MEMIT(1K)   | 40.89 | 44.98 | 38.18 | 40.07 |
> > > |             |NO EDITING(10K)| 30.28 | 30.54 | 29.68 | 30.65 |
> > > |             |  iReVa(10K)   | 51.89 | 99.98 | 79.06 | 28.44 |
> > >
> > >
> > > ## Question 3 (Multi-token objects)
> > >
> > > Thank you for your feedback; we now understand your concern. You are referring to whether, after editing a source question (denoted as `q`) + ["New", "Zealand"], some unrelated sentences ending with "New" (denoted as `p`) would mistakenly output "Zealand." For iReVa, avoiding this issue requires that the neurons corresponding to `q + "New"` are not activated, which depends on the model's encoding capabilities—that is, whether the model can distinguish between `q + "New"` and `p + "New"`.
> > >
> > > Since such datasets are hard to find, we will illustrate our approach with an example. Suppose we have three prompts, all ending with "New," but each followed by a different next word:
> > >
> > > - s1: "Wellington is located in New"
> > > - s2: "Ran Blake used to teach in New"
> > > - s3: "The biggest city in the US is New"
> > >
> > > Using GPT2-XL as the base model, the cosine similarity between these sentences is as follows:
> > > (s1, s2) → 0.7147, (s1, s3) → 0.6786, (s2, s3) → 0.6943.
> > >
> > > With GPT-J-6B as the base model:
> > > (s1, s2) → 0.5777, (s1, s3) → 0.6223, (s2, s3) → 0.6379.
> > >
> > > In iReVa, neurons are activated only when the similarity between the input sentence and the edited sentence exceeds a threshold, denoted as $\theta$. In our code, $\theta$ is set to 0.75 for GPT2-XL and 0.65 for GPT-J-6B, both higher than the pairwise similarities among the three sentences above. As a result, the model does not confuse them. Selecting $\theta$ based on such easily confusable examples is an effective way to choose this hyperparameter. Generally, larger models can better distinguish between these sentences (possibly due to larger hidden sizes or the model's awareness of the next token), leading to a lower $\theta$.
> > >
> > > ## Question 4 (Evaluation)
> > >
> > > ### Question 4.2 (CounterFact)
> > >
> > > We apologize for not clearly explaining the differences between the two datasets in the global rebuttal. Specifically, the training targets in the zsRE dataset `conflict` with the information the model learned during pretraining, so we refer to it as a *knowledge update*. In contrast, there is no such conflict in the PARAREL dataset, which we term as *knowledge insertion*. Additionally, the targets in PARAREL are shorter, with many being single-token targets, making it more challenging. For multi-token objects, we append the prefix of the target during testing to let the model predict the next token. These prefixes provide prior information that may allow the model to infer the next token, as you mentioned in Question 3.
> > >
> > > Regarding CounterFact, we believe its purpose is not to test the prediction accuracy of editing methods but to evaluate whether the model can *implicitly* increase the probability of the new target. The datasets we use, however, aim to test the model's ability to *explicitly* predict the new target. Therefore, we did not use CounterFact. Its main difference from zsRE lies in its paraphrase questions, which include much irrelevant information to mislead the model. Additionally, its neighborhood questions share the same next token as the source question, making it more confusing for autoregressive models.
> > >
> > > ### Question 4.3 (Paraphrase examples)
> > >
> > > The ripple effect you mentioned corresponds to what we referred to as *Portability* in the global rebuttal. A more detailed explanation can be found in Section 3 of the global rebuttal. Indeed, the ripple effect is a valuable capability worth studying. However, it is overly challenging for current editing methods. As we stated in the global rebuttal, due to the lack of interpretability in model reasoning, optimizing for the ripple effect often compromises the explainability of editing methods. Current methods (excluding prompting-based ones) struggle to make progress in this direction. Moreover, our experiments indicate that existing editing methods still have room for improvement in handling knowledge conflicts. Therefore, we did not evaluate the ripple effect metric.

---

> > > > ### Comment · Reviewer_WcNm · 2024-11-20
> > > >
> > > > I appreciate the authors' efforts to address my concerns and I would like to increase my score to a **6 (borderline accept)**. However, I don't think this work successfully addresses many of the limitations of existing knowledge editing methods. In my opinion, this is *"yet another knowledge editing technique"* that does better in some benchmarks, introducing new limitations of its own. Despite this, I think **iReVa** brings a new perspective to this very important research problem and is worth sharing with the community.
> > > >
> > > > I wish the authors good luck with their future work.

---

### Official Review · Reviewer_sgAZ · 2024-11-04

**Soundness:** 3
**Presentation:** 2
**Contribution:** 3
**Rating:** 5
**Confidence:** 3

**Summary:**

To address the high costs of fine-tuning in Knowledge Editing, it proposes a method called iReVa, which initializes and retrofits key-value pairs within MLP modules to construct new mappings of knowledge without affecting irrelevant information. Experiments show that iReVa outperforms existing methods in terms of edit success and generalization in two Knowledge editing benchmarks, and it also conducts the first knowledge withdrawal test.

**Strengths:**

* This paper presents a novel approach by expanding the original MLP's kv-pairs to store additional knowledge, thereby achieving knowledge updates. This idea is quite innovative.

* A new knowledge editing dataset is released.

* The method outperforms other major knowledge editing baselines on two benchmarks.

* The code is released.

**Weaknesses:**

1. L072-L073:  "In contrast, Meng et al. (2023a), through a cosine similarity analysis on hidden states experiment, posed viewpoints that the self-attention module can extract various types of knowledge". Is this a citation error? I don't believe the ROME paper conducted such an experiment. Please correct me if my understanding is incorrect.

2. It would be much more convincing if we could see some performance results on the LLaMA series models, such as LLaMA2-7B or LLaMA3-8B. Based on experience. Because knowledge editing methods tend to show varying performance differences when applied to LLaMA models.

3. L143-L146: Please double check the computation formulas inside the transformers block. Why is self-attention computed twice? It should only be computed once.

4. It would be helpful when the results include the "Probability" metric, which reflects whether the editing effects can cover other related knowledge. The details of this metric can be found in [1] and [2].

**Writing:**

(1) Please be careful of the \citep and \citet usage in the paper to make it more readable.

---

**References:**

[1] Evaluating the Ripple Effects of Knowledge Editing in Language Models

[2] Editing Large Language Models: Problems, Methods, and Opportunities

**Questions:**

Please see the Weaknesses section above.

---

> ### Author Response · Authors · 2024-11-19
>
> ### Reply to Reviewer sgAZ
>
> Thank you, Reviewer sgAZ, for your valuable feedback! We will address the weaknesses you highlighted in this reply.
>
> #### Weakness 1:
> We apologize for the citation error you mentioned. We intended to cite [1], and this issue will be corrected in the revised version of the paper.
>
> #### Weakness 2:
> We have already responded to this in Section 2 of the Author Global Rebuttal. If there is any remaining confusion, please feel free to contact us.
>
> #### Weakness 3:
> You mentioned that self-attention is computed twice in L143-L146. In fact, the first self-attention occurs in the module of the l-th layer, and the second occurs in the module of the (l+1)-th layer.
>
> #### Weakness 4:
> We believe you were referring to "Portability"? This metric evaluates reasoning problems associated with the edited knowledge. We have replied to this concern in Section 3 of the Author Global Rebuttal. If there are any additional questions, please let us know.
>
> #### Reference
>
> [1] Pmet: Precise model editing in a transformer

---

> > ### Author Response · Authors · 2024-11-24
> >
> > # Additional Relpy to reviewer sgAZ
> > As we approach the final day of discussion, we have noticed a lack of your engagement. We would greatly appreciate your assistance in coordinating the discussion. This reply will further elaborate on points from our previous responses.
> >
> > **Weakness 2** (Regarding the base model):
> > In the global rebuttal, we supplemented results for LLaMA3 on both our method and baselines. The experimental results demonstrate that our method remains advantageous, and this advantage becomes even more pronounced compared to GPT-J-6B. We believe it is necessary to explain the reason behind this phenomenon.
> >
> > For methods like MEMIT that update the model by writing update matrices into the weights, there is a significant issue with conflict during editing. Specifically, if the editing target conflicts with the knowledge acquired by the model during pretraining, the model may confuse `target_true` and `target_new` during inference, failing to produce the correct output. Fundamentally, this happens because such methods fail to precisely locate where the knowledge is stored within the model weights. For instance, if certain knowledge is edited into the 10th transformer layer while its actual storage location is in the 20th layer, during forward propagation, the model will retrieve `target_new` in the 10th layer and `target_true` in the 20th layer. This dual activation of memories associated with two pieces of knowledge causes confusion, leaving the model uncertain about which information to trust.
> >
> > In contrast, iReVa updates knowledge using an overwrite-based approach, allowing the model to rely more on the edited information. This conflict issue poses significant limitations in practical applications, making it quite challenging to update outdated or incorrect knowledge effectively.
> >
> > **Weakness 4** (Portability):
> > Another reason we did not evaluate portability is that existing methods are generally incapable of optimizing this aspect due to a lack of interpretability. As a result, comparisons on this metric hold little meaningful value. While this metric is undoubtedly a critical research direction for the future, we believe it would be more appropriate to explore it once we better understand the reasons behind multi-hop knowledge inference capabilities in models. Only then would further testing on this metric yield valuable insights.

---

### Official Review · Reviewer_21cm · 2024-11-09

**Soundness:** 2
**Presentation:** 3
**Contribution:** 3
**Rating:** 6
**Confidence:** 4

**Summary:**

This paper introduces an approach to editing knowledge in large language models called iReVa. The authors focus on the MLP blocks within Transformer modules, which they follow previous work and cast them as key knowledge carriers. Their method, iReVa, aims to insert new information into these blocks without disrupting existing knowledge. iReVa explicitly initializes and retrofits key-value pairs into MLP blocks to construct a new mapping of a piece of knowledge, aiming not damaging the irrelevant knowledge. The authors apply their approach on GPT-2, GPT-NEO and GPT-J models on two benchmark datasets, showing its potential in knowledge editing and maintaining the model's overall performance.

**Strengths:**

* The idea of iReVa is quite intuitive and straightforward. Un-edited data should keep their hidden states unchanged after knowledge editing, while edited data should have activation activated as expected.

* The results on two knowledge editing benchmark seem quite impressive, and especially the analysis of Figure 2, which compare baseline results of edits in various layers on zsRE dataset.

* The writing and idea are well presented. Figure 1 provides a very good and clear overview of iReVa.

**Weaknesses:**

* Although the authors run evaluations on three language models, namely GPT-2, GPT-NEO and GPT-J, these base models are not state-of-the-art any more. In addition, the evaluations are mainly for base models, where in real applications, practitioners may want to update their knowledge after fine-tuning with real world defeats feedbacks. Therefore, it will be interesting to see more results of LLaMA 3.1 models and their chat versions, as well.

* The knowledge editing tasks are somewhat too simple and target output seems quite short. Knowledge is a complex concept and a natural language sentence can include dense knowledge. For these two benchmarks used in this paper, their input prompts seem quite short. It is unclear that how this method is applicable in real world applications. For example, a new medical paper may have some new findings in their paper and how to use this paper' method inject these new knowledge into a medical language model?

* The generalization task evaluation is also not comprehensive. Although NQ dataset covers different types of knowledge, their scope is quite limited. It will be interesting to evaluate models on MMLU or MMLU-Pro benchmark data, which is much diverse and comprehensive than NQ dataset used in this paper.

* The multi-task object during knowledge editing involve multiple hyper-parameters for task balancing during training, which will introduce the complexity of tuning for specific domain and tasks.

**Questions:**

See comments in the Weaknesses.

---

> ### Author Response · Authors · 2024-11-19
>
> ### Reply to Reviewer 21cm
>
> Thank you, Reviewer 21cm, for your valuable feedback and recognition of our work! We will address the issues you mentioned under the weaknesses in this reply.
>
> #### Weakness 1 (Regarding the base model):
> We have already responded to this in Section 2 of the Author Global Rebuttal. If there is any remaining confusion, please feel free to contact us.
>
> #### Weakness 2 (The knowledge editing tasks are too simple):
> We acknowledge that our exploration of knowledge editing tasks is not yet exhaustive. Researchers do not fully understand the exact structure of knowledge storage in language models, and proposed theories (e.g., [1]) lack solid theoretical evidence. For instance, you mentioned injecting new discoveries from the medical domain into the model, which involves complex logical relationships. Currently, all editing methods struggle with such tasks due to limited understanding of how knowledge is stored in language models. Existing methods are based on hypotheses about this structure and attempt simpler editing tasks.
>
> In our work, we hypothesize that knowledge is stored as a (prefix, next token) pair: the prefix representation is stored in the first linear layer of the 2-layer feedforward network (FFN), and the next token's representation is stored in the second linear layer. Although this hypothesis may not be entirely accurate, it is intuitive and interpretable. Exploring more precise ways of knowledge storage remains an important direction for future research.
>
> #### Weakness 3 (Generalization task evaluation is not comprehensive):
> Indeed, in the zsRE dataset, the samples used to test Specificity (whether unrelated knowledge is affected by editing) come from NQ, a dataset proposed in [2] and widely used as a benchmark. You suggested using MMLU or MMLU-Pro for testing Specificity. However, Specificity only requires test samples unrelated to the edit input, so its diversity does not affect the metric.
>
> #### Weakness 4 (Multiple hyperparameters):
> iReVa does involve several parameters requiring manual adjustment. Apart from essential model training parameters, only two are noteworthy:
> 1. **Adaptor scale factor $\alpha$:** This requires exploration across multiple orders of magnitude.
> 2. **Activation bias $\theta$:** This is adjustable within the range [0,1], reducing tuning effort.
>
> Additionally, iReVa supports gradient-free insertion, significantly improving editing speed and simplifying hyperparameter tuning.
>
> #### Reference
>
> [1] Transformer Feed-Forward Layers Are Key-Value Memories
>
> [2] Fast model editing at scale

---

> > ### Comment · Reviewer_21cm · 2024-11-26
> >
> > Thank you for your response. I will keep my score as is.

---

### Author Response · Authors · 2024-11-19

# Author Global Rebuttal
We sincerely thank all the reviewers for their valuable feedback on our paper! This section serves as the global rebuttal to address common concerns raised by multiple reviewers.

## Comparison with Baselines
Reviewers WcNm and bY48 noted discrepancies between our reported results and those of MEMIT [1], particularly on the zsRE dataset (GPT-J-6B), where the performance of MEMIT in the original paper surpasses what we reported. After a thorough review of our implementation of MEMIT, we are confident there are no errors in our reproduction. However, there are differences in dataset preprocessing.

The zsRE dataset includes a question requiring editing and two possible answers: a factual (ground-truth) answer and a new conflicting answer. Our approach, iReVa, uses the new conflicting answer as the training target, whereas MEMIT uses the ground-truth answer, as evidenced by their source code (/dsets/zsre.py). Subsequent works also follow this setup. Specifically, when using MEMIT’s setup with the ground-truth answer, we can easily reproduce results close to those reported in MEMIT’s original paper. However, when using the new answer as the training target, the results align with those reported in our paper.

The reason for adopting this setup is based on the goal of knowledge editing task, which involves inserting or updating knowledge in the model. Training with the ground-truth answer supports insertion but not updating, as the model can answer some zsRE questions correctly by existing knowledge obtained in pre-training phase.

For our proposed PARAREL dataset, the answers are also factual ground-truth answers. However, given that we use GPT-2 XL (1.5B) as the backbone, the model can hardly to answer most questions in the dataset. This allows PARAREL to evaluate knowledge insertion capabilities effectively. For larger backbone models, many questions might be answerable by the model itself, leaving fewer instances to test the editing method’s insertion ability.

## Using LlAMA3 as the Backbone
Reviewers 21cm and sgAZ requested results for iReVa on the LlAMA3 backbone. Our paper does not include these results because we believe LlAMA models have an architecture incompatible with some baseline methods like MEMIT. MEMIT assumes a 2-layer feedforward network (FFN) structure in transformer MLPs, which LlAMA models do not use. Instead, LlAMA’s MLP contains three trainable layers $U,D,G$, with forward propagation defined as $y=[f(xG) \otimes (xU)]D$ (where $\otimes$ denotes the Hadamard product).

After a discussion with reviewers, we found that MEMIT operates on the second MLP within the two-layer FFN (referred to as $K$ and $V$ later). Specifically, it modifies the $V$ matrix. Even within the LlAMA series models, the corresponding Down matrix $D$ can also be regarded as serving the same function as $V$.

Moreover, iReVa is applicable to almost any computational model, including LlAMA3. Below are iReVa results on zsRE-10K using LlAMA3. Considering the time overhead, we only tested 1K examples for MEMIT. Results show that iReVa with 10K edits outperforms MEMIT with 1K edits, and we suggest that MEMIT on LlAMA3 underperforms MEMIT on GPT-J-6B due to more knowledge conflicts.
|  Backbone   |    Method     | S $\uparrow$  | ES $\uparrow$ | PS $\uparrow$ | NS $\uparrow$ |
| :---------: | :-----------: | :---: | :---: | :---: | :---: |
|             | NO EDITING(1K)| 35.99 | 32.36 | 31.12 | 49.19 |
| LlAMA3.1-8B |   MEMIT(1K)   | 40.89 | 44.98 | 38.18 | 40.07 |
|             |NO EDITING(10K)| 30.28 | 30.54 | 29.68 | 30.65 |
|             |  iReVa(10K)   | 51.89 | 99.98 | 79.06 | 28.44 |

## Portability in Reasoning Tasks
Reviewers sgAZ and bY48 raised questions related to portability (reasoning ability) of editing method. Reasoning has always been a challenging issue in the field of model editing. In the background of large language models, reasoning ability lacks interpretability, making it difficult for most model editing methods, including iReVa, to apply newly learned knowledge in reasoning tasks. One existing attempt is IKE [2], which inspired from in-context learning. However, this approach affects interpretability and locality metrics.

To enable iReVa to handle reasoning tasks, one possible solution could involve approaches like Chain-of-Thought (CoT) [3]. These methods guide the model to answer intermediate questions related to edits before generating a full response to the reasoning problem step by step. Overall, the reasoning ability of current model editing methods often trades off with interpretability, which many researchers are striving to solve.

## Reference
[1] Mass-Editing Memory in a Transformer

[2] Can we edit factual knowledge by in-context learning?

[3] Chain-of-thought prompting elicits reasoning in large language models

---

### Author Response · Authors · 2024-11-22

Dear Reviewer,

Thank you for your comments. Since the discussion deadline is approaching, could you please have a look at our rebuttal and give us some feedbacks? Your responses will be highly appreciated. Thank you.

Best,

Authors

---

### Meta-Review · Area_Chair_R8dd · 2024-12-21

**Metareview:**

This paper presents iReVa for editing knowledge in large language models (LLMs). Building on prior work, the authors identify MLP blocks within Transformer modules as key knowledge carriers. The approach focuses on inserting new information into these blocks while preserving existing knowledge. Experiments on GPT-2, GPT-NEO, and GPT-J using two benchmark datasets demonstrate iReVa's potential for effective knowledge editing while maintaining overall model performance.

While the reviewers found the idea of iReVa is quite intuitive and straightforward with good presentations, there are several major weaknesses:
1. Method design might be impractical. The paper proposes to add the adaptor to the original model, however, some oracle information is used in this model, i.e., this method needs to let the model know which is the final token. This is impractical in real world.
2. Experimental results are not convincing. For example, the authors did not use the deduplicated dataest from MEMIT on zsRE-10k, leading to unfair comparisons. There is doubt on if the implementation is correct and the hyper-parameter is properly tuned. The ideal case would be evaluating iReVa on the exactly same dataset used in MEMIT.
3. The knowledge editing tasks are somewhat too simple and target output seems quite short. Knowledge is a complex concept and a natural language sentence can include dense knowledge. For these two benchmarks used in this paper, their input prompts seem quite short. It is unclear that how this method is applicable in real world applications.
4. The generalization task evaluation is not comprehensive. Although NQ dataset covers different types of knowledge, their scope is quite limited. It will be interesting to evaluate models on MMLU or MMLU-Pro benchmark data, which is much diverse and comprehensive than NQ dataset used in this paper.

Although the authors addressed some of the questions in their rebuttal, several major concerns remains unsolved. Therefore, the paper is not ready to be published at its current form.

**Additional Comments On Reviewer Discussion:**

Although the authors addressed some of the questions in their rebuttal, several major concerns remains unsolved (see above). Therefore, the paper is not ready to be published at its current form.

---

### Decision · Program_Chairs · 2025-01-22

Reject